# Solution of the Drug Resistance Problem of *Escherichia coli* with Silver Nanoparticles: Efflux Effect and Susceptibility to 31 Antibiotics

**DOI:** 10.3390/nano13061088

**Published:** 2023-03-17

**Authors:** Ekaterina Nefedova, Nikolay N. Shkil, Nikolay A. Shkil, Diana Garibo, Roberto Luna Vazquez-Gomez, Alexey Pestryakov, Nina Bogdanchikova

**Affiliations:** 1Siberian Federal Scientific Centre of Agro-BioTechnologies of the Russian Academy of Sciences, Novosibirsk 630501, Russia; 2Research School of Chemistry and Applied Biomedical Sciences, Tomsk Polytechnic University, Tomsk 634050, Russia; 3Consejo Nacional de Ciencia y Tecnología, Ciudad de México 03940, Mexico; 4Centro de Nanociencias y Nanotecnología, Universidad Nacional Autónoma de México, Ensenada 22800, Baja California, Mexico; 5Escuela de Ciencias de la Salud, Universidad Autónoma de Baja California, Ensenada 22890, Baja California, Mexico

**Keywords:** medicine resistance, *Escherichia coli*, AgNPs, efflux effect

## Abstract

The current work is a continuation of our studies focused on the application of nanoparticles of metallic silver (AgNPs) to address the global problem of antibiotic resistance. In vivo, fieldwork was carried out with 200 breeding cows with serous mastitis. Ex vivo analyses showed that after the cow was treated with an antibiotic-containing drug Dienomast^TM^, *E. coli* sensibility to 31 antibiotics decreased by 27.3%, but after treatment with AgNPs, it increased by 21.2%. This could be explained by the 8.9% increase in the portion of isolates showing an efflux effect after Dienomast^TM^ treatment, while treatment with Argovit-C^TM^ resulted in a 16.0% drop. We verified the likeness of these results with our previous ones on *S. aureus* and *Str. dysgalactiae* isolates from mastitis cows processed with antibiotic-containing medicines and Argovit-C^TM^ AgNPs. The obtained results contribute to the recent struggle to restore the efficiency of antibiotics and to preserve the wide range of antibiotics on the world market.

## 1. Introduction

In recent decades, antibiotics are used to treat infectious diseases. However, some bacteria have evolved to become resistant to these antibiotics. The resistance of bacteria to antibiotics has become a serious problem worldwide. It is predicted that by 2050, around 10 million people will die from infections due to antibiotic-resistant bacteria [1]. Antibiotic resistance also has negative impacts on healthcare costs. Recent reports estimate that antibiotic resistance may cost from USD 300 billion to more than USD 1 trillion annually by 2050 worldwide [2]. Therefore, the World Health Organization has proclaimed antibiotic resistance as one of the priority concerns in healthcare [3]. Several alternatives to solve the worldwide problem of AMR have been described: (1) organic compounds (phytochemicals, essential oils, metabolites from medicinal plants, etc.); (2) biomolecules (antibodies, bacteriocins, enzymes, etc.); (3) phage therapy; and (4) physicochemical methods (photoinactivation, chemotherapy, nonthermal atmospheric pressure plasmas, etc.) [4]. Although organic molecules have antimicrobial activity, they are expensive and have low stability [4]. Phage therapy and physicochemical methods are laborious and require complex equipment and skilled workers. So, the task of developing low-cost and effective antimicrobial agents capable to overcome antimicrobial resistance is still relevant. Currently, nanomaterials have received great attention as novel antimicrobials [5]. Silver nanoparticles (AgNPs) are the most studied nanoparticles in biomedical applications due to their antibacterial, antifungal, antiviral, and anticancer properties [6]. So, AgNPs application as one of the promising alternatives for AMR combating is one of the nowadays lines of investigation.

Antibiotic resistance extends from edible animals to humans via the food chain [7]. Cows top the list of edible animals due to the high volume of products derived from them and consumed by humans (meat, milk, cheese, yogurt, sour cream, cottage cheese, etc.). In the dairy industry, mastitis is one of the most widespread diseases, which requires expensive treatment and causes a capital loss of 124 EUR/cow/year, leading to EUR 125 billion losses worldwide [8].

In our previous works, we showed that the treatment of mastitis cows with Argovit-C^TM^ AgNPs increased the susceptibility of *Staphylococcus aureus* and *Streptococcus dysgalactiae* to 31 antibiotics by 11% and 13%, respectively, while treatment with antibiotic-containing drugs decreased their susceptibility by 25% and 23%, respectively [9,10]. These effects were attributed to the capability of Argovit-C^TM^ AgNPs to reduce the ratio of isolates having an efflux effect, although treatments with antibiotic-containing drugs led to an increase in this ratio [9,10]. The present study is the continuation of our studies carried out with two Gram-positive bacteria *S. aureus* and *Str. dysgalactiae* [9,10], and its aim is to find out if the same effects are characteristic of Gram-negative bacteria such as *E. coli*.

## 2. Materials and Methods

### 2.1. Experimental Procedure

This research was carried out on 200 breeding farm cows having serous mastitis. The experimental protocol was approved by the Ethical Committee of the Federal State Budgetary Institution of the Siberian Federal Scientific Center for Agrobiotechnologies of the Russian Academy of Sciences (Decision No. 00017 on 10 February 2017). The animals used in this study were properly cared for by humans. The diagnosis as well as complete recovery were determined based on clinical symptoms and confirmed using a Biochemical California test [11]. The cows were partitioned into 2 groups of 100 animals each (Figure 1). Group I included cows treated with Dienomast^TM^, a first-line veterinary medicine used for the treatment of mastitis, and group II included 100 cows treated with Argovit-C^TM^. The milk samples from these groups were analyzed before and after intracisternal treatments with Dienomast^TM^ (group I) or Argovit-C^TM^ (group II).

### 2.2. Sampling

The milk samples were taken from mastitis cows before and after treatment with Dienomast^TM^ or Argovit-C^TM^ in livestock farms in the Novosibirsk region during 2019–2020. For sampling, the udder teats were wiped with a cotton swab moistened with 70° ethanol. Then, 10 mL of milk was collected in sterile tubes, avoiding contact between the nipple and the tube’s edge. The test tube with milk was closed with a cotton-gauze stopper, and the name or inventory number of the cow was recorded on the tube label. The samples were kept at 8–10 °C until testing and delivered for research within 3–4 h.

### 2.3. Treatment Formulations

Dienomast^TM^ (CJSC NPP Agrofarm, Voronezh, Russia) is an antibacterial medicine recommended for intracisternal introduction containing 8.75 mg of dioxidine and 17.5 mg of gentamicin sulfate as active ingredients. In veterinary medicine, Dienomast^TM^ is used as a first-line drug for mastitis treatment. Cows were intracisternally injected with Dienomast^TM^ at a dose of 5 mL with an interval of 24 h. The treatment was finished after 6 days, when complete recovery occurred.

Argovit-C^TM^ (Research and Production Center “Vector-Vita”, Novosibirsk, Russia) was kindly provided by Dr. Vasily Burmistrov. It is applied for prophylactic and therapeutic purposes in cases of gastrointestinal diseases of calves. Argovit-C™ is a suspension of silver nanoparticles (AgNPs) at a concentration of 200 mg/mL (20% by weight). The content of metallic silver is 12 mg/mL (1.2 wt.%). The size of the silver metallic particles is in the range of 5–20 nm with an average diameter of 15 nm. AgNPs are stabilized with polyvinylpyrrolidone and collagen hydrolysate with a total content of 18.8 and 80 wt.% of distilled water. Argovit-C^TM^ at 10-fold dilution (1.2 mg/mL of metallic Ag) was administered intracisternally to cows of group II at a dose of 10 mL per day.

### 2.4. Isolation and Identification of E. coli

Endo’s medium was used to isolate *E. coli* from milk before and after the treatment of cows with Dienomast^TM^ or Argovit-C^TM^. The identification of the microbiota isolated from milk was carried out by taking into account the morphological, biochemical, and cultural properties of the bacteria according to accepted methods [12].

### 2.5. Antimicrobial Sensitivity and Efflux Effect Testing

The sensitivity of *E. coli* was measured using a kit designed for testing the susceptibility of *E. coli* to antimicrobial drugs, which contained biochemical plates that differentiate enterobacteria (OOO NPO Diagnostic Systems, Nizhny Novgorod) according to the criterion of the European Committee for Antimicrobial Susceptibility Testing [13].

The efflux-effect test consisted of *E. coli* culture seeding on Eugonic agar with ethidium bromide (1 mg/L). After 24 h of cultivation, the efflux effects in cultures were analyzed using a transilluminator. In the case of efflux-effect absence, ethidium bromide penetrated inside the bacterial cell and remained inside the cell, forming a complex with DNA, which glowed under ultraviolet light, whereas fluorescence absence indicated that ethidium bromide was ejected, revealing the presence of efflux.

The susceptibility and efflux studies of *E. coli* isolated from the milk were carried out ex vivo for 31 antibiotics from the following 8 groups: (1) penicillins (carbenicillin, ampicillin, oxacillin, benzylpenicillin (penicillin), and amoxicillin); (2) tetracyclines (tetracycline and doxycycline); (3) macrolides (erythromycin and tylosin); (4) aminoglycosides (amikacin, neomycin, streptomycin, kanamycin, gentamicin, and tobramycin); (5) cephalosporins (cephalexin, cefuroxime, cefotaxime, ceftiofur, cefazolin, and ceftazidime); (6) fluoroquinolones (ciprofloxacin, enrofloxacin, norfloxacin, and ofloxacin); (7) nitrofurans (furagin and furazolidone); (8) others (lincomycin, polymyxin, rifampicin, and levomycetin (chloramphenicol)). These studies were carried out before and after treatment with the drugs.

### 2.6. Ag Concentration in Milk and Blood

The measurements were carried out using atomic absorption spectroscopy with a Spectrophotometer AA-7000 (Shimadzu, Japan).

### 2.7. Statistical Analysis

The obtained results were statistically processed with the methods of parametric and nonparametric analyses. Statistical analysis was performed with STATISTICA 13.3 software. The accumulation, correction, and systematization of initial information, as well as visualization of the results, were carried out using GraphPad Software 9.0 (San Diego, CA, USA).

## 3. Results

### 3.1. Physicochemical Properties of Argovit-C^TM^ AgNPs

The physicochemical properties of Argovit-C^TM^ AgNPs were described in [14]. Shortly, Argovit-C^TM^ AgNPs are spherical particles with an average diameter of 14.95 [± 10.1] nm, particle size in the range of 1–55 nm, hydrodynamical diameters with two average sizes (44 and 164 nm), the broad plasmon resonance peak at 420–460 nm, and zeta potential of 9.6 [±0.6] mV (Figure 2).

### 3.2. Treatment Duration

The duration of treatments till complete recovery with the two formulations were 6.2 ± 0.2 and 3.2 ± 0.3 days, for Dienomast^TM^ and Argovit-C^TM^, respectively.

### 3.3. Isolation and Identification of Bacteria

The results of the analysis of the microbial community’s contribution in milk samples taken from 200 cows with mastitis are shown in Table 1. The isolate contributions were 140 isolates (70%) of *E. coli*; 20 isolates (10%) of *S. aureus*; 12 isolates (6%) of *Str. dysgalactiae*; 12 isolates (6%) of *Staphylococcus epidermidis*; 10 isolates (5%) of *Streptococcus agalactiae*; and 6 isolates (3%) of *Streptococcus pyogenes* (Table 1).

In the present work, the results for *E. coli* are presented, and then they are compared with the results obtained for *S. aureus* and *Str. dysgalactiae* published in [9,10], respectively. The results for the bacteria of other types will be presented elsewhere. Of the 200 milk samples (from both groups I and II) analyzed before treatment, a total of 140 *E. coli* isolates were obtained (77 in group I and 63 in group II). As expected, fewer *E. coli* isolates were obtained after the treatments for both groups, with 100 isolates (62 in group I and 38 in group II).

After the isolation of bacteria and Gram staining, *E. coli* species were identified via isolates’ biochemical characterization. The results are presented in Table 2.

### 3.4. Antibiotic Susceptibility Changes after Treatments

The bactericidal effect of 31 antibiotics on *E. coli* isolates before and after cow treatment with Dienomast^TM^ and Argovit-C^TM^ (with and without the efflux effect) are shown in Figure 3. To make these differences more visible, these results are shown in Figure 4a,b as the percentage difference between susceptibility to 31 antibiotics before and after Dienomast^TM^ and Argovit-C^TM^ treatments, respectively. The contribution of *E. coli* isolates before and after the treatments are presented in Figure 4c,d for Dienomast^TM^ and Argovit-C^TM^, respectively. All graphs in Figure 4 include the results for isolates with and without the efflux effect. A significant result was the observation that Dienomast^TM^ treatment led to a notable susceptibility drop, while Argovit-C^TM^ treatment led to an increase in antibiotic susceptibility (Table 3 and Table 4, Figure 3 and Figure 4).

### 3.5. Dienomast^TM^

The *E. coli* isolates were not sensitive to four antibiotics (tobramycin, penicillin, furagin, and furazolidone) before and after Dienomast^TM^ treatment. After Dienomast^TM^ treatment, the susceptibility of *E. coli* decreased by 24.4% and 30.2% for the isolates with and without the efflux effect, respectively. After Dienomast^TM^ treatment, susceptibility disappeared toward four antibiotics (neomycin, doxycycline, cefotaxime, and erythromycin) for the isolates with efflux and toward six antibiotics (neomycin, norfloxacin, oxacillin, amoxicillin, cefotaxime, and erythromycin) for the isolates without the efflux effect (Figure 3a and Table 3). The susceptibility toward two and one antibiotics slightly increased (data above 0%) after Dienomast^TM^ treatment for the *E. coli* isolates without and with the efflux effect, respectively (Figure 4a and Table 3). For the majority of the cases, susceptibility toward antibiotics decreased after Dienomast^TM^ treatment (Figure 3a,c and Figure 4a; Table 3).

### 3.6. Argovit-C^TM^

After Argovit-C^TM^ treatment, *E. coli* susceptibility toward antibiotics increased (for isolates with and without the efflux effect) by 21.2% on average. Thus, the susceptibility difference was positive (the interval of activity difference between +0% and +100%, Figure 4b and Table 4). The activity diminished only by 2.0% for the isolates with the efflux effect for a unique antibiotic named enrofloxacin (Figure 4b and Table 4). The *E. coli* isolates with and without the efflux effect were not susceptible to penicillin and furazolidone before and after Argovit-C^TM^ treatment (Figure 4b).

### 3.7. Changes in the Contribution of Isolates with Efflux Effect after Treatments

Changes in the contribution of *E. coli* isolates with the efflux effect after treatment with Dienomast^TM^ and Argovit-C^TM^ are shown in Figure 3c,d, respectively.

### 3.8. Ag Concentration in Milk and Blood

The silver concentration in milk was measured before AgNP administration and 6, 12, and 24 h after each administration, and 48, 72, and 96 h after the last administration. The quantified Ag concentration was not more than 33.5–39.2 µg/L. According to the Agency for Toxic Substances and Disease, Ag concentration in potable water in the USA is up to 80 µg/L [15]. Hence, the measured Ag concentration in cow milk was two times less than in drinking water. It should also be noted that silver was not detected in cow blood after AgNP treatment. This indicates that the applied AgNP treatment does not lead to AgNP penetration into the blood and then into cow organs.

## 4. Discussion

After Dienomast^TM^ treatment, the sensitivity of *E. coli* toward 31 antibiotics decreased by an average of 27.3%. By contrast, after Argovit-C^TM^ treatment, the susceptibility of *E. coli* to 31 antibiotics increased by an average of 21.2% (Table 2 and Table 3). Figure 4a,b show a decrease in susceptibility to antibiotics, which occurred after treatment with DienomastTM (Figure 4a), and an increase in susceptibility to antibiotics, which occurred after treatment with Argovit-CTM (Figure 4b). It can be concluded that the total gain in the use of Argovit-C^TM^ instead of Dienomast^TM^ in terms of the antibiotic resistance of *E. coli* in mastitis treatment was 48.5%. It should be noted that only for four and two antibiotics for Dienomast^TM^ and Argovit-C^TM^, respectively, exceptions were observed: A change in susceptibility was in the opposite direction (Table 3 and Table 4). Moreover, susceptibility remained absent for eight isolates after Dienomast^TM^ treatment and only for four isolates after Argovit-C^TM^ treatment (Table 3 and Table 4). Moreover, susceptibility completely disappeared for 10 isolates after Dienomast^TM^ treatment, and none disappeared after Argovit-C^TM^ treatment (Table 3 and Table 4).

The portion of isolates showing the efflux effect grew by 8.9% after Dienomast^TM^ treatment and, by contrast, dropped by 16.0% after Argovit-C^TM^ treatment. It can be concluded that the total gain in the use of Argovit-C^TM^ instead of Dienomast^TM^ in terms of the capacity to eliminate antibiotics from *E. coli* bacteria was 24.9%. Thus, the ability to eliminate antibiotics from *E. coli* bacteria was enhanced after Dienomast^TM^ treatment and decreased after Argovit-C^TM^ treatment.

These results are very similar to those observed in our previous works devoted to an analogous study of (1) *S. aureus* in the treatment of cow mastitis with Lactobay^TM^ and Argovit-C^TM^ [9] and (2) *Str. dysgalactiae* in the treatment of cow mastitis with Spectromast^TM^ LC and Argovit-C^TM^ [10]. Lactobay^TM^, Spectromast^TM^ LC, and Dienomast^TM^ are first-line drugs based on antibiotics for the treatment of cow mastitis. The chemical structures of the antibiotic drugs used in the present study and our previous works [9,10] are presented in Figure 5.

Figure 6c,d present a comparison of the results of two previously studied bacteria, i.e., *S. aureus* (blue symbols) [9] and *Str. dysgalactiae* (red symbols) [10], and *E. coli* (green symbols) for antibiotic drugs Lactobay^TM^, Spectromast^TM^ LC, and Dienomast^TM^ (a and c) and for Argovit-C^TM^ (b and d).

The results of this comparison indicated that the treatment of serous mastitis with antibiotic-containing drugs (Lactobay^TM^, Spectromast^TM^ LC, and Dienomast^TM^) led to a 25.1, 22.9, and 27.3% fall in *S. aureus*, *Str. dysgalactiae*, and *E. coli* sensitivity toward the 31 antibiotics, respectively (Figure 6a), whereas after Argovit-C^TM^ treatment, *S. aureus*, *Str. dysgalactiae*, and *E. coli* sensitivity toward the antibiotics increased by 11.4, 13.1, and 19.4%, respectively (Figure 6b). It can be concluded that with the use of Argovit-CTM instead of antibiotic-containing drugs, the total decrease in antibiotic resistance of *S. aureus*, *Str. dysgalactiae*, and *E. coli* was 36.5, 26.0 and 48.5%, respectively.

The portion of isolates showing the efflux effect for *S. aureus*, *Str. dysgalactiae*, and *E. coli* increased by 16.1, 7.5, and 8.9% after Lactobay^TM^, Spectromast^TM^ LC, and Dienomast^TM^, respectively (Figure 6c). In comparison, the portion of isolates showing the efflux effect for these bacteria decreased by 18.6, 17.8, and 16.0 % after Argovit-C^TM^ treatment (Figure 6d). The number of antibiotics for which the sensitivity of *S. aureus*, *Str. dysgalactiae*, and *E. coli* remained absent or disappeared after Lactobay^TM^, Spectromast^TM^ LC, and Dienomast^TM^ treatments (for isolates with and without the efflux effect) were 19, 10, and 18, respectively, and after all 3 Argovit-C^TM^ treatments, they were 3–6 times less (3, 3, and 4, respectively) (Figure 6 and [9,10]). All these results for the three bacteria indicate that the results for the two Gram-positive bacteria *S. aureus* and *Str. dysgalactiae* and the Gram-negative bacterium *E. coli* have similar tendencies.

The revealed phenomenon that the capacity to eliminate antibiotics from the three studied bacteria increased after treatments with antibiotic-containing drugs and decreased after Argovit-C^TM^ treatment can be at least partially explained by the effect of an increase in resistance to antibiotics after treatment with antibiotic-containing drugs, and the effect of a decrease in resistance to antibiotics after treatment with Argovit-C^TM^. In [16], it was revealed that under the influence of Ag+ and AgNPs, the re-sensitization of multidrug-resistant (mcr-1)-positive *E. coli* to colistin (cationic polypeptide antibiotic) occurred through binding and functional disruption of MCR enzymes. Further experiments are needed for the investigation of the mechanism of re-sensitization of *E. coli* to 31 different antibiotics after cow treatment with AgNPs.

Therefore, the finding of this study indicating that the efflux effect of antibiotics from the three bacteria (*S. aureus*, *Str. dysgalactiae*, and *E. coli*) increased after treatments with antibiotic-containing drugs and decreased after Argovit-CTM treatment can explain at least partially the observed effect of an increase in resistance to antibiotics after treatment with antibiotic-containing drugs, and the effect of a decrease in resistance to antibiotics after treatment with Argovit-CTM.

Additionally, it was revealed that complete recovery from cow serous mastitis occurred after 6 days of Lactobay^TM^ treatment and 4 days of Argovit-C^TM^ treatment [9]; after 5 and 3 days of Spectromast^TM^ LC and Argovit-C^TM^ treatments, respectively, and after 6 days of Dienomast^TM^ treatment [10]; and after 3 days of Argovit-C^TM^ treatment (this work). Hence, recovery after treatment with Argovit-C^TM^ occurred 33–50% faster than that after taking antibiotic-containing drugs.

So, Argovit-C^TM^ treatment is more effective than treatment with Lactobay^TM^, Spectromast^TM^ LC, and Dienomast^TM^ treatments from three perspectives. Argovit-C^TM^ (1) reduces antibiotic resistance, (2) decreases bacteria’s capacity to eliminate antibiotics from inside their cells, and (3) accelerates the rate of cow recovery. Another advantage of Argovit-C^TM^ is that since the year 2000, it was certified for veterinary use as a prophylactic and therapeutic drug for calve gastrointestinal diseases. This will simplify the certification procedure of Argovit-C^TM^ for cow mastitis treatment.

In the literature, we did not find any results concerning the influence of AgNPs on the susceptibility of bacteria to antibiotics and on the efflux effect of bacteria in vivo aside from our previous articles about the study of *S. aureus* [9] and *Str. dysgalactiae* [10]. The experiments described in the literature were carried out in vitro [9,10], while the present work is the first translational research study including an in vivo field study with 200–400 breeding farm cows and detecting the rise in the sensitivity of 3 mastitis-causing bacteria to 31 antibiotics after treatments with AgNPs.

In future in vivo studies, we plan to investigate whether the decrease in resistance to antibiotics after AgNP treatment can be not only observed for *S. aureus*, *Str. dysgalactiae*, and *E.coli* but also for other bacteria, and not only for mastitis but also for other diseases.

## 5. Conclusions

The present work is the continuation of our previous translational research with the application of Argovit-C^TM^ AgNPs as an approach to solving the worldwide problem of drug resistance. Our previous two works were dedicated to the study of *S. aureus* and *Str. dysgalactiae* [9,10]. The present work was dedicated to cow mastitis treatment with Argovit-C^TM^ AgNPs and the first-line antibiotic drug Dienomast^TM^ regarding the susceptibility of *E. coli* (which is a Gram-negative bacterium, while *S. aureus* and *Str. dysgalactiae* are Gram-positive). The results for all three bacteria were very similar:The treatment of mastitis with antibiotic-containing medicines (Lactobay^TM^, Spectromast^TM^ LC, and Dienomast^TM^) led to a 23–27% fall in *S. aureus*, *Str. dysgalactiae*, and *E. coli* sensitivity toward 31 antibiotics, while after Argovit-C^TM^ treatment, their susceptibility toward the antibiotics increased by 11–19.4%.The total number of antibiotics for which the susceptibility of these three bacteria remained absent or disappeared after treatments with antibiotic-containing drugs was between 10 and 19, while for Argovit-C^TM^ treatments, this number was 3 and 4.The portion of isolates showing the efflux effect for these three bacteria increased by 8–16% after treatment with antibiotic-containing drugs, while it decreased by 16–19% after Argovit-C^TM^ treatment. The changes observed in susceptibility after treatments with Argovit-C^TM^ and antibiotic-containing drugs can be at least partially explained by the alteration in the contribution of isolates with the efflux effect after treatments.Mastitis recovery with Argovit-C^TM^ use occurred 33% to 50% faster than that with antibiotic-containing drugs.

To the best of our knowledge, our translational research, performed in our previous studies [9,10] and the present work, is the first in vivo fieldwork performed with 200–400 breeding farm cows that shows the possibility of decreasing the resistance of *S. aureus*, *Str. dysgalactiae*, and *E. coli* isolates to 31 antibiotics. The results of this work contribute to the current struggle to recover the activity of antibiotics and preserve a wide range of antibiotics in the market.

## Figures and Tables

**Figure 1 nanomaterials-13-01088-f001:**
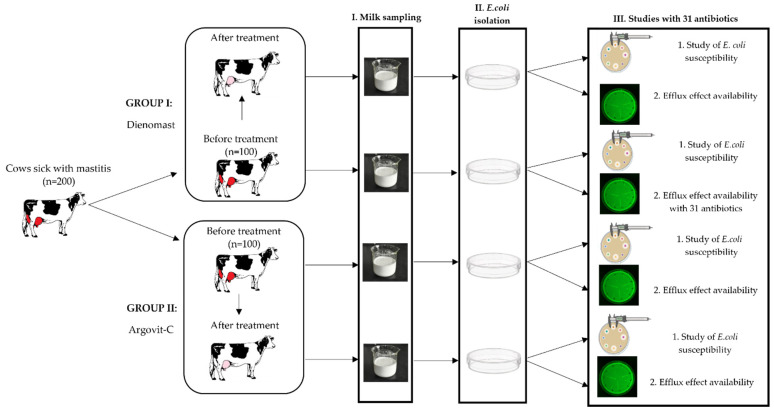
Diagram of experimental design for *E. coli* isolates before and after treatments with Dienomast^TM^ and Argovit-C^TM^: stage I—milk sampling, stage II—isolation of *E. coli*, and stage III—studies of the sensitivity of *E. coli* to 31 antibiotics and the presence of efflux effect.

**Figure 2 nanomaterials-13-01088-f002:**
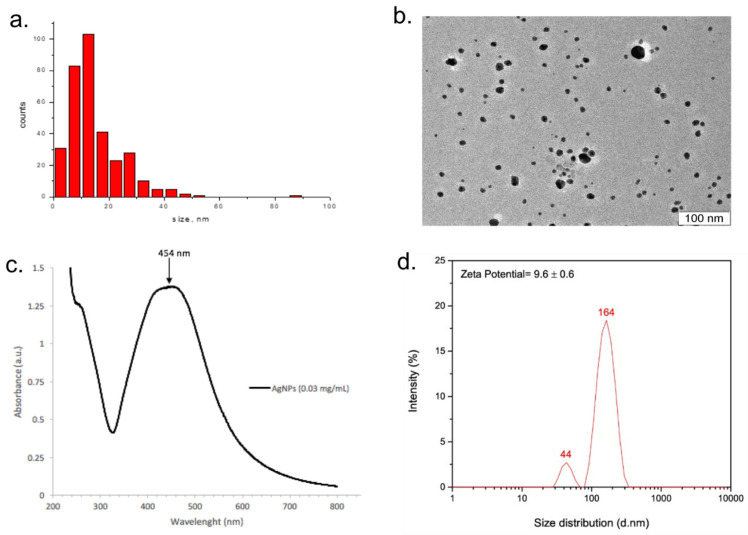
Physicochemical characteristics of Argovit-C^TM^ AgNPs. TEM data: (**a**) AgNP size distribution, (**b**) TEM image, (**c**) UV–visible absorption spectrum (for 0.03 mg/mL of metallic silver), and (**d**) zeta potential and hydrodynamic diameter. The figure is taken from [14].

**Figure 3 nanomaterials-13-01088-f003:**
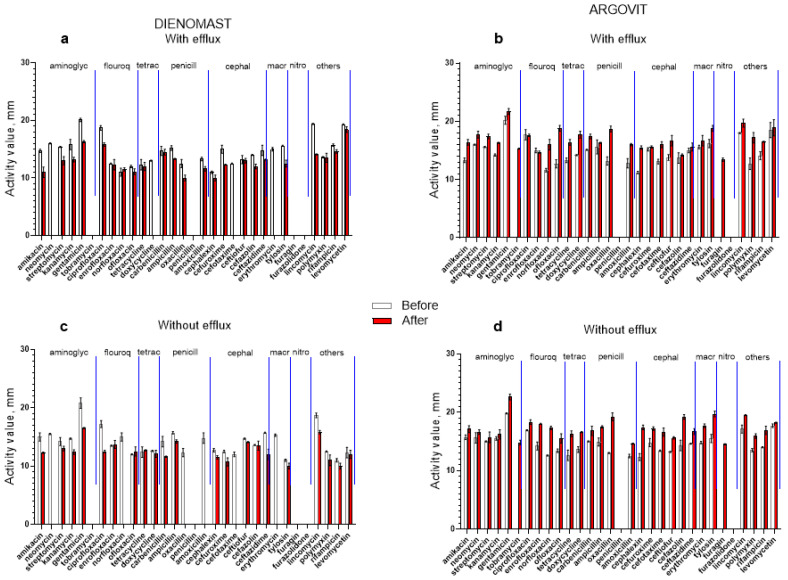
Susceptibility of *E. coli* to 31 antibiotics before (white columns) and after (red columns) treatment with Dienomast^TM^ (**a**,**c**) and Argovit-C^TM^ (**b**,**d**). Isolates with (**a**,**b**) and without (**c**,**d**) efflux effect.

**Figure 4 nanomaterials-13-01088-f004:**
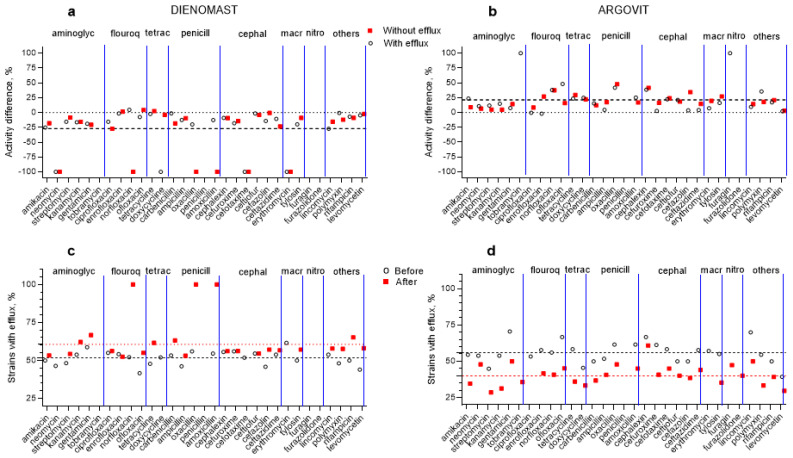
Percentage susceptibility difference (susceptibility after treatment minus susceptibility before treatment) of *E. coli* to 31 antibiotics: (**a**) Dienomast^TM^ and (**b**) Argovit-C^TM^ treatments; isolates with (circles) and without (squares) an efflux effect. Contribution of isolates having an efflux effect for (**c**) Dienomast^TM^ and (**d**) Argovit-C^TM^ treatments: circles indicate before treatments, and squares indicate after treatments.

**Figure 5 nanomaterials-13-01088-f005:**
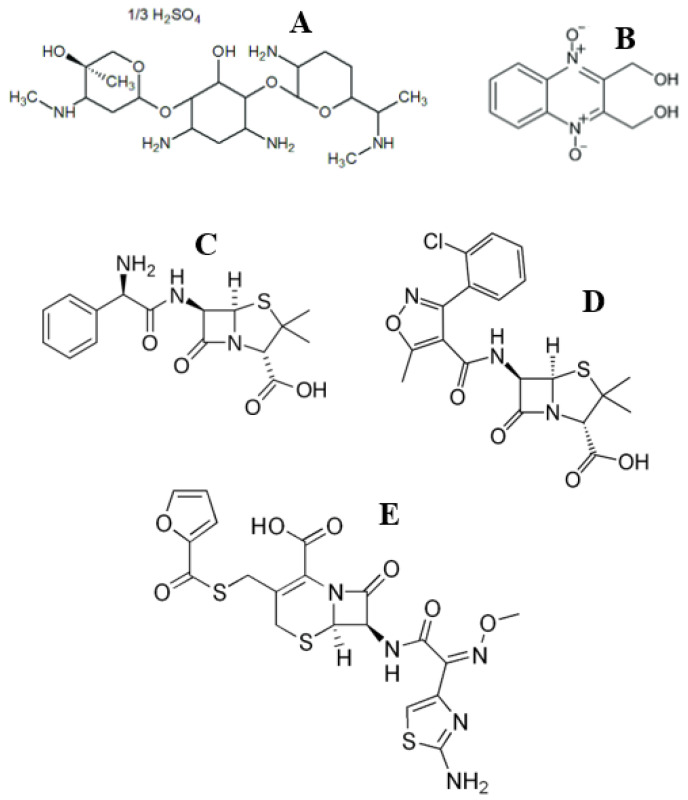
Chemical structures of antibiotic drugs used in the present work: Dienomast^TM^: gentamicin (**A**) and dioxidine sulfate (**B**) and in the previous works [9,10]: Lactobay^TM^: ampicillin (**C**) and cloxacillin (**D**) and Spectromast^TM^ LC: ceftiofur (**E**).

**Figure 6 nanomaterials-13-01088-f006:**
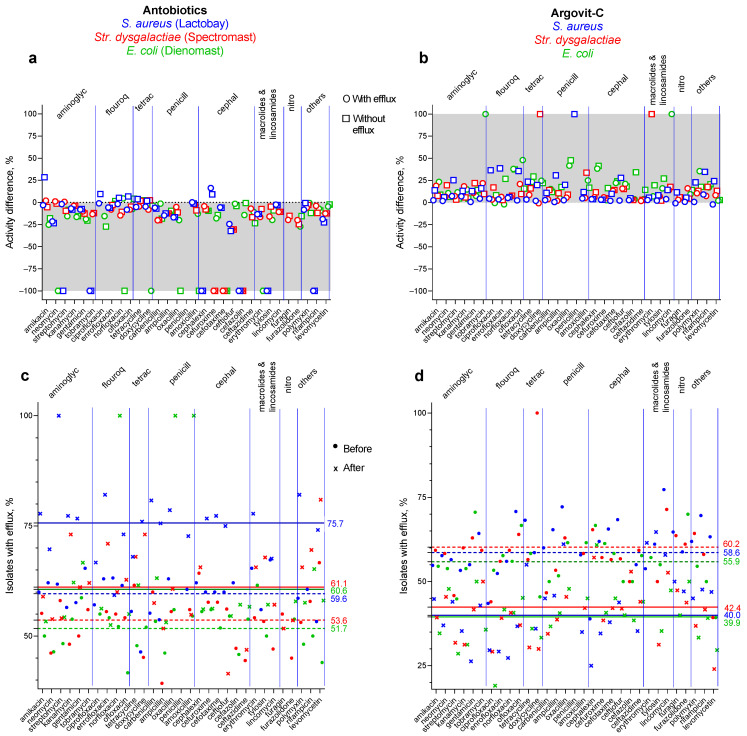
Results of susceptibility and portion of isolates showing efflux for three bacteria: *S. aureus* (blue symbols) data acquired from our previous work [9], *Str. dysgalactiae* (red symbols) data acquired from our previous work [10], and *E. coli* (green symbols), the present work, for antibiotic drugs Lactobay^TM^, Spectromast^TM^ LC, and Dienomast^TM^ (**a**,**c**) and for Argovit-C^TM^ (**b**,**d**): (**a**,**b**) susceptibility data for isolates with efflux effect (circles) and without efflux effect (squares); (**c**,**d**) portion of isolates showing efflux effect before (circles) and after (x) cow treatments.

**Table 1 nanomaterials-13-01088-t001:** Results of the microbiological study of milk samples from 200 cows with serous mastitis.

Microorganisms	Contribution of Isolates with Specific Microorganisms in 200 Milk Samples	
	Number	Percentage, %
*E. coli*	140	70
*S. aureus*	20	10
*Str. dysgalactiae*	12	6
*S. epidermidis*	12	6
*Str. agalactiae*	10	5
*Str. pyogenes*	6	3

**Table 2 nanomaterials-13-01088-t002:** Results of the biochemical test of *E. coli* isolates.

Parameter	*E. coli*
Sodium citrate	-
Mannose	-
Sodium citrate with glucose	+
Arginine	-
Indole	+
Reaction Voges–Proskauer	-
Ornithine	+
Urea	-
H_2_S	-
Phenylalanine	-
Lysine	+
Glucose	+
Lactose	+
β-Galactosidase	+
Sodium malonate	+
Sucrose	+
Inositol	-
Sorbitol	+
Arabinose	+
Maltose	+
Temperature	37 °C

+ means that *E. coli* utilizes the indicated compound and - means that *E. coli* does not utilize the indicated compound.

**Table 3 nanomaterials-13-01088-t003:** Change in *E. coli* susceptibility to antibiotics (for isolates with and without efflux effect) caused by Dienomast^TM^ treatment.

	Isolates without Efflux Effect			Isolates with Efflux Effect	
Activity Change Classification	Number of Antibiotics	Average Change in Activity		Number of Antibiotics	Average Change in Activity
Cumulative change					
Activity variation	27	−30.2 %		25	−24.4%
Total activity variation for 52 samples is −27.3%					
		Details of changes			
Activity remained absent	4	0		4	0
Activity disappeared (−100%)	6	−100%		4	−100%
Activity appeared (+100%)	0	0		0	0
Activity decreased (−∆%)	18	−12.3%		22	−12.0%
Activity increased (+∆%)	3	+2.7%		1	+4.5%
Activity remained constant (0%)	0	0		0	0

**Table 4 nanomaterials-13-01088-t004:** Change in *E. coli* susceptibility to antibiotics (for isolates with and without efflux effect) caused by Argovit-C^TM^ treatment.

	Isolates without Efflux Effect			Isolates with Efflux Effect	
Activity Change Classification	Number of Antibiotics	Average Change in Activity		Number of Antibiotics	Average Change in Activity
Cumulative change					
Activity change	29	+19.4%		29	+22.9%
Total activity variation for 59 samples is +21.2%					
		Changes detailed			
Activity remained absent	2	0		2	0
Activity disappeared (−100%)	0	0		0	0
Activity appeared (+100%)	2	+100%		2	+100%
Activity decreased (−∆%)	0	0		2	−1.3 %
Activity increased (+∆%)	27	+19.4%		25	+19.4%
Activity remained constant (0%)	0	0		0	0

## Data Availability

Data are available upon request from the corresponding author.

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
