# Peer review of "Solution of the Drug Resistance Problem of Escherichia coli with Silver Nanoparticles: Efflux Effect and Susceptibility to 31 Antibiotics"

_nanomaterials, 2023, doi:10.3390/nano13061088_

Round 1
Reviewer 1 Report
In this work, authors presented their work on the in-vivo study of anti-serous mastitis effects using Ag-NPs based Argovit-C and antibacterial Dienomast. The results show that treatment with Ag-NPs could increase the E. coli susceptibility towards antibiotics while with antibiotic containing medicine the susceptibility dropped. In general the experiments are well planned and the findings interesting.
Below are some comments/questions I have for the authors:
1. Format-wise, there’s some slight issue with the line numbers starting with 180.
2. Since the ultimate goal is to avoid antibiotic resistance from edible animal to human, in this study via milk, is there any test run on the metabolism pathway for AgNPs? Will any of the registered NPs be detected in the milk?
3. Is there any reason why figure 4 (data completely from previous work) is included in this manuscript?
4. Could you include discussions on the mechanism of improved antibiotics susceptibility with Ag-NPs?
5. Please proof read the manuscript again as grammar issues are here and there. Some of the expressions include: line 66 grow/growth; line 72, line 76 care/cared; line 298 and 315.
Author Response
Please, find an attached file

Reviewer 2 Report
The manuscript by Garibo and coworkers describe the use of dienomast and Argovit-C to treat mastitis in cattle. This work is an extension of their previous work, however, E. coli was used this time. The research design is good and the data is solid. Nonetheless, there are major grammatical errors that need to be addressed. Also, there are some statements that are a little bit concerning, such as "to solve the global problem of antibiotic resistance". This study can not do this, especially by analyzing the data presented. It is true that some promising results were obtained but claiming that this will solve the issue is wrong. In summary, the idea is good, the data and number of samples are good, the effects observed are low, and the novelty of this work is low; but the presentation quality is terrible (English). This manuscript needs a major English edition. The manuscript may be considered after major editions by a native speaker.
Author Response
Please, find an attached file

Reviewer 3 Report
Ekaterina Nefedova et al presented Ag NPs against antibiotic resistance of Escherichia coli. This work was organized well. Experimental data strongly support their conclusions. Basically, I think it could be accepted for publication in this journal after considering the following points.
1. The authors should give SEM or TEM images, size distribution, surface modification of AgNPs.
2. If possible, the chemical structures of antibiotic drugs for comparison also are required.
3. To improve the quality, a graphic abstract is suggested to add.
4. What is the mechanism of antibiotic resistance in this work?
5. The authors explain the possible advantages and disadvantages of this approach, compared to existing others.
6. Some updating reports on metallic nanoparticles and bioapplications are recommended to be cited, such as ACS Nano, 2014, 8, 8529-8536; ACS Nano 2018, 12, 11139; Proc. Natl. Acad. Sci. U. S. A. 2022, 119, e2119417119.
Author Response
Please, find an attached file

Round 2
Reviewer 1 Report
Authors have successfully addressed my concerns from the previous round. I believe the discussion in the manuscript is more complete to be published.
However I would still suggest authors to have the manuscript proofread for grammar: for example the first sentence "During decades the antibiotics are used to treating infectious diseases" should be "During decades antibiotics are used to treat infectious diseases."
Reviewer 2 Report
The authors have addressed the comments.